EMBO
reports

# scientific report

# Chemerin15 inhibits neutrophil-mediated vascular inflammation and myocardial ischemia-reperfusion injury through ChemR23

*Jenna L. Cash[†+], Stefania Bena\*, Sarah E. Headland\*, Simon McArthur, Vincenzo Brancaleone & Mauro Perretti*
William Harvey Research Institute, Barts and the London School of Medicine and Dentistry, London

**Neutrophil activation and adhesion must be tightly controlled to prevent complications associated with excessive inflammatory responses. The role of the anti-inflammatory peptide chemerin15 (C15) and the receptor ChemR23 in neutrophil physiology is unknown. Here, we report that ChemR23 is expressed in neutrophil granules and rapidly upregulated upon neutrophil activation. C15 inhibits integrin activation and clustering, reducing neutrophil adhesion and chemotaxis *in vitro*. In the inflamed microvasculature, C15 rapidly modulates neutrophil physiology inducing adherent cell detachment from the inflamed endothelium, while reducing neutrophil recruitment and heart damage in a murine myocardial infarction model. These effects are mediated through ChemR23. We identify the C15/ChemR23 pathway as a new regulator and thus therapeutic target in neutrophil-driven pathologies.**

Keywords: chemerin peptide; inflammation; intravital microscopy; neutrophil; resolution

## INTRODUCTION

In vertebrates, tissue trauma, infection and ischaemia–reperfusion injury prompt rapid neutrophil extravasation from the circulation to the site of injury [1]. Neutrophil trafficking follows a multi-step cascade of leukocyte–endothelial interactions initiated upon capture of the circulating cell by selectins. Neutrophil rolling, also mediated by selectins, follows, enabling the leukocyte to interact with chemokines exposed on the endothelium. Inside-out signalling triggered by these chemokines or cytokines leads to neutrophil β2 integrin activation, which involves a conformation

change from a bent, low affinity form to an extended, high affinity conformation. Integrin pro-adhesive activity can also be increased by clustering, whereby the integrin accumulates in discrete areas of the plasma membrane [2]. Interaction of these activated β2 integrins with their respective ligands results in neutrophil adhesion, followed by intravascular crawling. Neutrophils then undergo transendothelial migration and move along chemotactic gradients towards the inflammatory site [3,4].

Knockout mice, neutralizing antibodies and the existence of pathological conditions such as leukocyte adhesion deficiency, have underlined the importance of integrins, integrin activation and CD62L (L-selectin) in neutrophil recruitment [4,5]. Excessive intravascular neutrophil recruitment and ensuing activation is a key pathogenic feature of numerous vascular diseases, including ischaemia–reperfusion injury; for example, post-myocardial infarction [6] and atherosclerosis [7]. It is clear that a delicate balance must exist to ensure effective removal of the inciting inflammatory insult, while avoiding overly aggressive or prolonged inflammatory responses that are detrimental to the host. However, little information exists regarding inhibitors of neutrophil integrin activation and thus recruitment during inflammation.

Higher organisms have evolved a network of anti-inflammatory and pro-resolving pathways, which counter-regulate inflammatory responses and promote regain of tissue homeostasis, ensuring that the inflammatory response is limited in magnitude, time and space. Improved understanding of these endogenous regulatory systems could pave the way for the development of therapeutic approaches to tame inflammatory pathologies [8,9]. Recently, we identified a novel pathway for inflammatory resolution, whereby chemerin15 (C15), a 15-aa peptide derived from the chemoattractant protein chemerin, inhibits pro-inflammatory mediator production by macrophages and promote phagocytosis of apoptotic cells through the receptor ChemR23 [10,11].

ChemR23 was originally detected on monocytes, macrophages and dendritic cells [12,13]; however, granulocyte expression has not been investigated. In the present study, we provide the first compelling evidence that ChemR23 is expressed on neutrophils and can be harnessed by the pro-resolving peptide C15 to restrict excessive neutrophil trafficking to inflammatory loci, including in

William Harvey Research Institute, Barts and the London School of Medicine and Dentistry, Charterhouse Square, London EC1M 6BQ,
*These authors contributed equally to this work.
†Present address: Physiology & Pharmacology, Medical Sciences Building, University of Bristol, Bristol, UK.
+Corresponding author. Tel: +44 117 331 2209; Fax: +44 117 331 2288; E-mail: jenna.cash@bristol.ac.uk or j.cash@qmul.ac.uk

Received 19 March 2013; revised 30 July 2013; accepted 2 August 2013; published online 3 Septermber 2013

a murine myocardial infarct model. C15 appears to achieve these effects by fine-tuning neutrophil–endothelial cell interactions through modulation of β2 integrin activation (affinity) and clustering (avidity) and L-selectin shedding.

## RESULTS
### ChemR23 is rapidly upregulated on neutrophils

Previous studies have focused on the role of chemerin and its receptor ChemR23 on dendritic cells, monocytes and macrophages [10–13], revealing important modulatory actions. Using flow cytometry we found that ChemR23 is also expressed on 68% of resting human neutrophils (Fig 1A,B). Neutrophil ChemR23 expression rapidly increased upon stimulation with pro-inflammatory mediators, tumor necrosis factor alpha (TNFα), fMLF and interleukin-8 or exposure of isolated neutrophils to activated endothelial cells under flow, but was unaffected by anti-inflammatory mediators (annexin A1; α-melanocyte-stimulating hormone; or C15; Fig 1A–C). We also demonstrate that ChemR23 is expressed on circulating murine neutrophils from wild-type but not ChemR23$^{-/-}$ mice (Fig 1D). As ChemR23 surface expression is upregulated within minutes, we suspected expression within neutrophil granules which can undergo rapid mobilization to fuse with the cells plasma membrane. Congruently, immunofluorescence studies revealed punctate intracellular ChemR23 expression on resting human neutrophils, a distribution, which became polarized upon cell activation with TNFα (Fig 1E). Staining for neutrophil granule markers suggests ChemR23 colocalization with neutrophil secretory vesicle marker CD35 and specific granule marker CD66b, but not azurophil granule marker CD63 (Fig 1F). Rapid modulation of ChemR23 expression through degranulation, as shown for the pro-resolving mediator annexin A1, could provide a mechanism to quickly alter neutrophil phenotype, even in the vasculature, allowing the cell to perceive and respond to signals from the environment in a manner that is dependent upon the particular vesicles or granules mobilized [14].

As neutrophil ChemR23 expression has not been documented before, we sought to verify the presence of the functional GPCR using calcium mobilization assays. C15 and chemerin, but not C15-S (scrambled C15 peptide), induced calcium flux responses in human neutrophils, which could be inhibited with the ChemR23 antagonist CCX2005 (Fig 1G). The expression of functional ChemR23 on human and murine neutrophils opens the possibility that ChemR23 ligands could directly modulate neutrophil reactivity.

### Chemerin15 selectively modulates neutrophil physiology

To assess the potential impact of C15 on neutrophil physiology, we investigated its effect on CD62L (L-selectin) and PSGL-1, which are implicated in mediating neutrophil rolling on the vascular endothelium. In particular, CD62L shedding results in higher neutrophil rolling velocities and is thought to reduce its cross-linking thus limiting neutrophil β2 integrin activation [15,16]. Treatment of human resting neutrophils with 10 pM C15 resulted in 50% shedding of CD62L and a 25% reduction in PSGL-1 expression (Fig 2A, supplementary Fig S1A online), suggesting that C15 might have a role in regulating neutrophil rolling events. Other anti-inflammatory molecules known to suppress neutrophil activation and induce L-selectin shedding are annexin A1 and non-steroidal anti-inflammatory drugs [17,18].

We next determined the effect of C15 on β2 integrin activation and clustering, which are key events in increasing the pro-adhesive activity of the integrin and thus are critical to neutrophil adhesion, intravascular crawling and extravasation. Integrin activation can be assessed using antibodies that specifically detect the extended high affinity conformation of the integrin. Neutrophil pre-treatment with C15 prior to stimulation with TNFα led to significant inhibition of CD18 (60%), CD11b (75%) and CD11a (58%) activation (Fig 2B, supplementary Fig S1B online). However, C15 was unable to modulate total CD11b levels (supplementary Fig S2A online) or neutrophil degranulation (supplementary Fig S2B online), indicating that C15 specifically interferes with integrin activation rather than degranulation-mediated upregulation of integrin expression. We next assessed the effect of C15 on CD11a and CD11b clustering (avidity), whereby integrin can accumulate in discrete areas of the plasma membrane [2]. Using fluorescence microscopy, we found that the relatively dispersed distribution of CD11a and CD11b in vehicle-treated neutrophils became much more clustered upon TNFα treatment and that this response was profoundly reduced by C15 (Fig 2C).

We next assessed whether the observed effects of C15 on integrin activation and clustering could affect neutrophil binding and adhesion to the β2 integrin ligand ICAM-1. Indeed, we found that C15 inhibits human neutrophil adhesion and spreading to immobilized ICAM-1 by 65% (Fig 2D, representative micrographs in Fig 2F). Furthermore, ChemR23 antagonism using CCX2005 (100 nM) significantly attenuated C15-elicited suppression of ICAM-1 adhesion (Fig 2E,F) and CD11b activation (Fig 2G). In agreement with these results, wild-type but not ChemR23$^{-/-}$ murine neutrophils treated with C15 show marked reduction in binding of soluble ICAM-1-Fc chimeric protein (Fig 2H).

To determine whether C15 can regulate integrin-dependent neutrophil chemotaxis *in vitro*, we used live cell tracking of neutrophils adherent to ICAM-1-coated IBIDI μ-slides and then treated with C15 in the presence of a fMLF gradient. C15 significantly impaired neutrophil chemotaxis (representative plots shown in Fig 2I), quantified by measuring centre of mass (spatial averaged point of all cell endpoints) an indicator of cell directionality and velocity (Fig 2J).

In the above assays, we have focused on the effect of C15 on neutrophil β2 integrin activation and downstream events as the role of β2 integrins in neutrophil physiology and neutrophil-driven inflammatory pathologies is widely appreciated. However, neutrophils also express β1 integrin (CD29 [19]) which, though less extensively elucidated, also appears to have a role in mediating neutrophil adhesion. We found that C15 suppresses β1 integrin activation (supplementary Fig S3A online) and adhesion to β1 integrin ligand fibronectin in wild-type but not ChemR23$^{-/-}$ neutrophils (supplementary Fig S3B online). C15 also inhibits β2 and β1 integrin activation on other ChemR23$^+$ cell types (for example, monocytes; supplementary Fig S4A,B online) and induced by other ligands (for example, fMLF; supplementary Fig S4C online) suggesting that to some extent the effects of C15 on integrin activation are independent of integrin subtype, cell type (providing ChemR23 is expressed) and integrin activator. C15 might achieve these effects by impinging on a common component of the inside-out signalling pathway that elicits integrin activation.

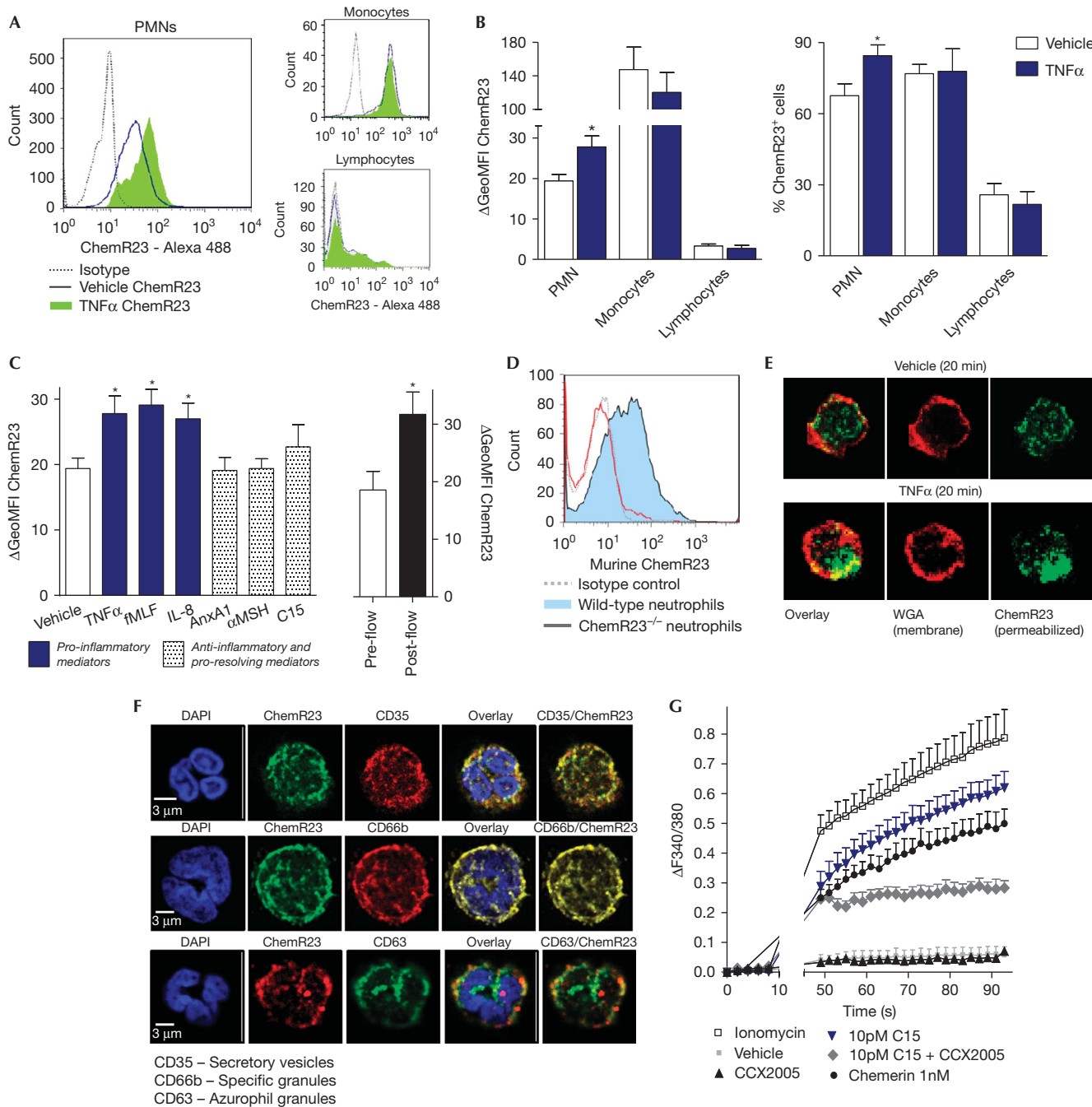

Fig 1 | Non-genomic modulation of ChemR23 expression on neutrophils (A,B) ChemR23 expression on human leukocytes was determined by flow cytometry on vehicle and TNFα (10 ng/ml; 20 min)-stimulated cells. Representative histograms shown in A. (C) Neutrophils were treated with vehicle, pro-inflammatory (TNFα; fMLF, 1 μM; IL-8, 100 ng/ml) or anti-inflammatory mediators (annexin A1, 10 nM; αMSH, 10 nM; C15, 10 pM) followed by staining for ChemR23. ChemR23 expression was also assessed on isolated human neutrophils before and after flow over activated endothelial cells. (D) ChemR23 expression by wild-type but not ChemR23$^{-/-}$ murine neutrophils. (E) Permeabilized human neutrophils were stained with anti-ChemR23 (with goat-anti-mouse Alexa 488 secondary) and wheat germ agglutinin (WGA-Alexa 647) to visualize the cell membrane. Cells were analysed by confocal microscopy. (F) Human neutrophils were stained for ChemR23 and markers of secretory vesicles (CD35), specific granules (CD66b) and azurophil granules (CD63). (G) Neutrophils were loaded with Fura2-AM and calcium flux responses elicited by control (ionomycin), vehicle, ChemR23 inhibitor (CCX2005, 100 nM), C15 (10 pM), scrambled C15 (C15-S; 10 pM) or chemerin (1 nM). Responses are displayed as ΔF340/F380. Graphs show mean values ± s.e.m. from three to six independent experiments. *$P < 0.05$ relative to vehicle-treated or pre-flow cells. αMSH, alpha-melanocyte-stimulating hormone; IL-8, interleukin-8; TNFα, tumour necrosis factor alpha.

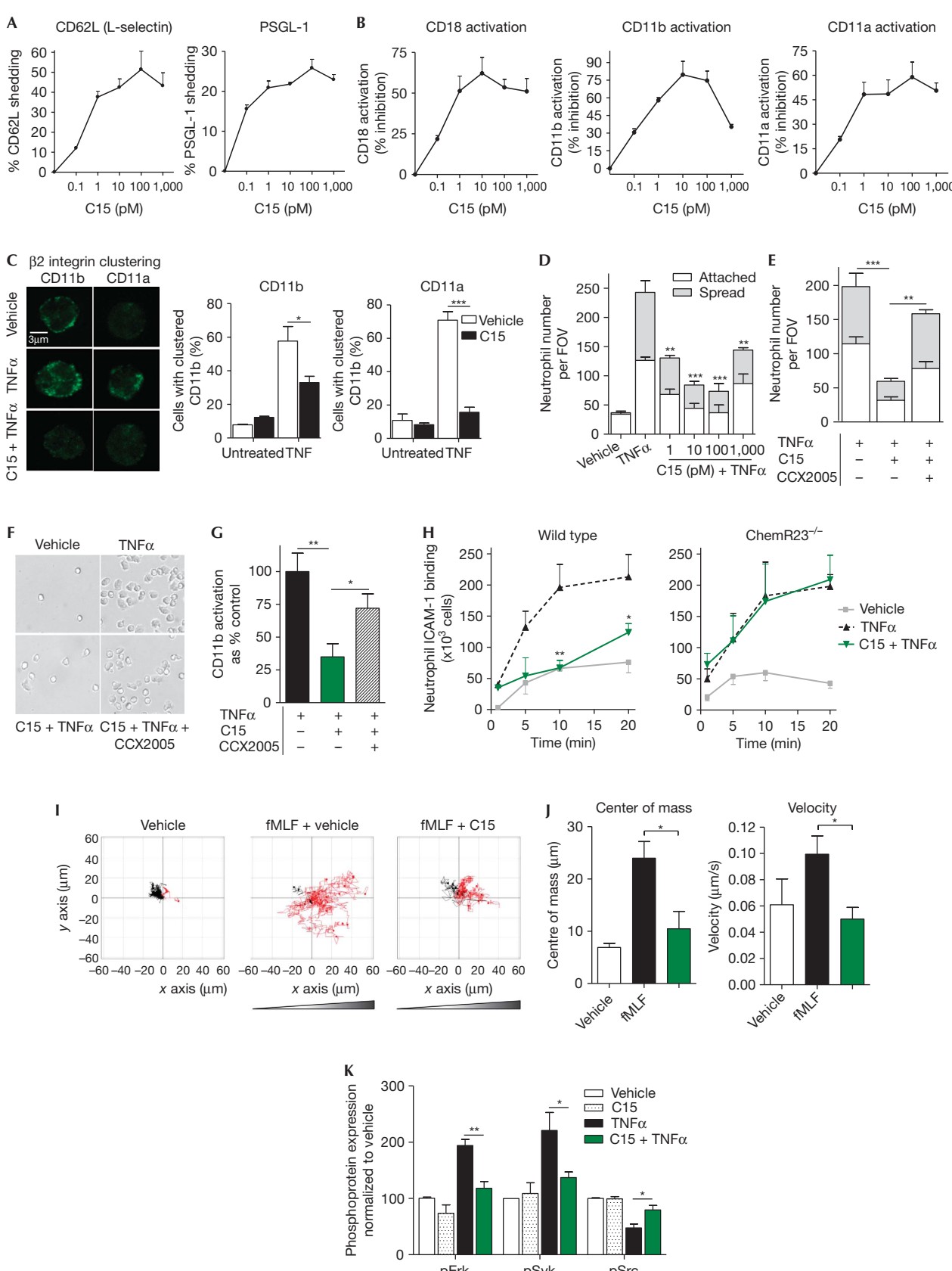

◄ **Fig 2** | C15 peptide selectively modulates neutrophil physiology through ChemR23 (**A**) Cells were treated with C15 (0.1–1,000 pM, 20 min), stained with anti-CD62L and anti-PSGL-1 and marker expression analysed by flow cytometry. (**B**) Cells were treated with C15 (0.1–1,000 pM, 10 min) and then stimulated with TNFα (10 ng/ml, 20 min). Cells were stained for activated β2 integrins; CD18, CD11b and CD11a and expression analysed by flow cytometry. (**C**) Cells were treated with vehicle or C15 (10 pM, 10 min) and then stimulated with TNFα (10 min). Integrin clustering was assessed by staining cells for CD11a and CD11b and analysing distribution by fluorescence microscopy. (**D**) Cells were treated with vehicle or C15 (10 pM, 10 min) and allowed to adhere to ICAM-1-coated multispot slides in the presence of TNFα for 10 min in static conditions. (**E**) Effect of ChemR23 antagonist (CCX2005) on C15-mediated inhibition of neutrophil static adhesion. (**F**) Representative micrographs of static adhesion assay are shown. (**G**) Effect of ChemR23 antagonist on C15-mediated inhibition of CD11b activation. (**H**) Wild-type and ChemR23$^{-/-}$ mouse neutrophils were pre-treated with vehicle or C15 (10 pM) followed by exposure to soluble ICAM-1-Fc in the presence or absence of TNFα for 1, 5, 10 or 20 min. ICAM-1 binding was quantified by staining with F'ab-FITC and analysing by flow cytometry. (**I**) Human neutrophils were pre-treated with vehicle or C15 (100 pM) then allowed to adhere to ICAM-1-coated IBIDI μ-slides. Chemotaxis towards fMLF was then monitored in real time over 30 min. Representative trajectory paths of 30 randomly chosen cells are shown. Red paths indicate rightward movement and black paths leftward movement. (**J**) Quantification of integrin-dependent neutrophil chemotaxis. Centre of mass (spatial averaged point of all cell endpoints) and velocity. (**K**) Neutrophils were pre-treated with C15 (10 pM) or vehicle prior to administration of vehicle or TNFα (10 min). Intracellular signalling flow cytometry was then performed following staining for pErk, pSyk and pSrc. Data are expressed as means ± s.e.m. from four to seven independent experiments; *$P < 0.05$; **$P < 0.01$; ***$P < 0.001$ relative to TNFα or fMLF-treated cells (**D,E,G,H,J,K**) or vehicle-treated cells (**C**) unless otherwise noted. C15, chemerin 15; FOV, field of view. ICAM, intracellular adhesion molecule; TNFα, tumour necrosis factor alpha.

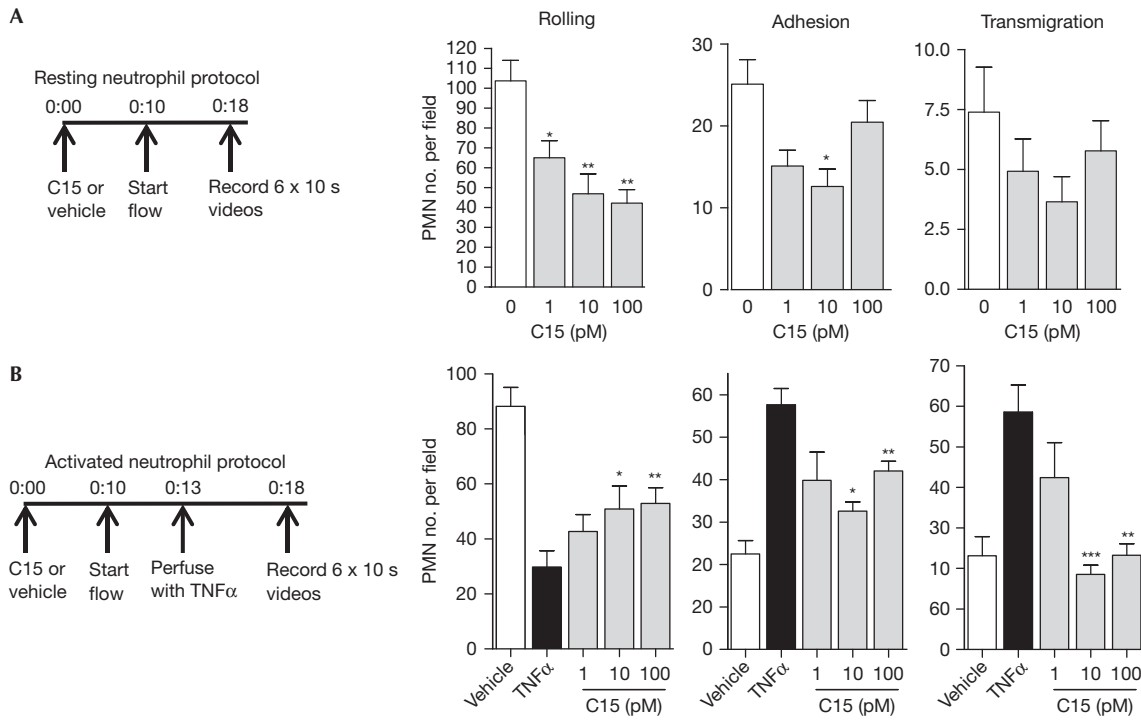

**Fig 3** | C15 potently inhibits human neutrophil–endothelial cell interactions under shear conditions (**A**) Resting neutrophils—cells were pre-treated with vehicle or C15 (1–100 pM) for 10 min and then perfused over TNFα-treated HUVECs. (**B**) Activated neutrophils—cells were treated with vehicle or C15 (1–100 pM) for 10 min and perfused with TNFα to induce neutrophil activation while flowing over TNFα-treated HUVECs. Neutrophil–endothelial cell interactions were classified as rolling, adhesion or transmigration. Data are expressed as means ± s.e.m. from six independent experiments. *$P < 0.05$; **$P < 0.01$; ***$P < 0.001$ relative to vehicle-treated cells (**A**) or TNFα-treated cells (**B**). HUVEC, human umbilical vein endothelial cells; T, TNFα-treated; TNFα. tumour necrosis factor alpha.

We therefore assessed the effect of C15 on common components of inside-out signalling leading to β2 integrin activation and clustering, Syk, Erk and Src kinases [20–24]. We used an intracellular flow cytometry approach to assess the phosphorylation status of these enzymes. pSyk (Y525/526) and pERK (Y202/204) represent activated enzymes while de-phosphorylation of Y527 of Src results in its activation. Neutrophil stimulation with TNFα activated all three enzymes, whereas neutrophil pre-treatment with 10 pM C15 prevented TNFα-induced activation of Syk (69%), ERK (80%) and Src (61%; Fig 2K).

Collectively, these data indicate that C15 engages ChemR23, inhibits neutrophil adhesion and chemotaxis by interfering with integrin activation and clustering by impinging on common components of the inside-out signalling pathway, thus affecting

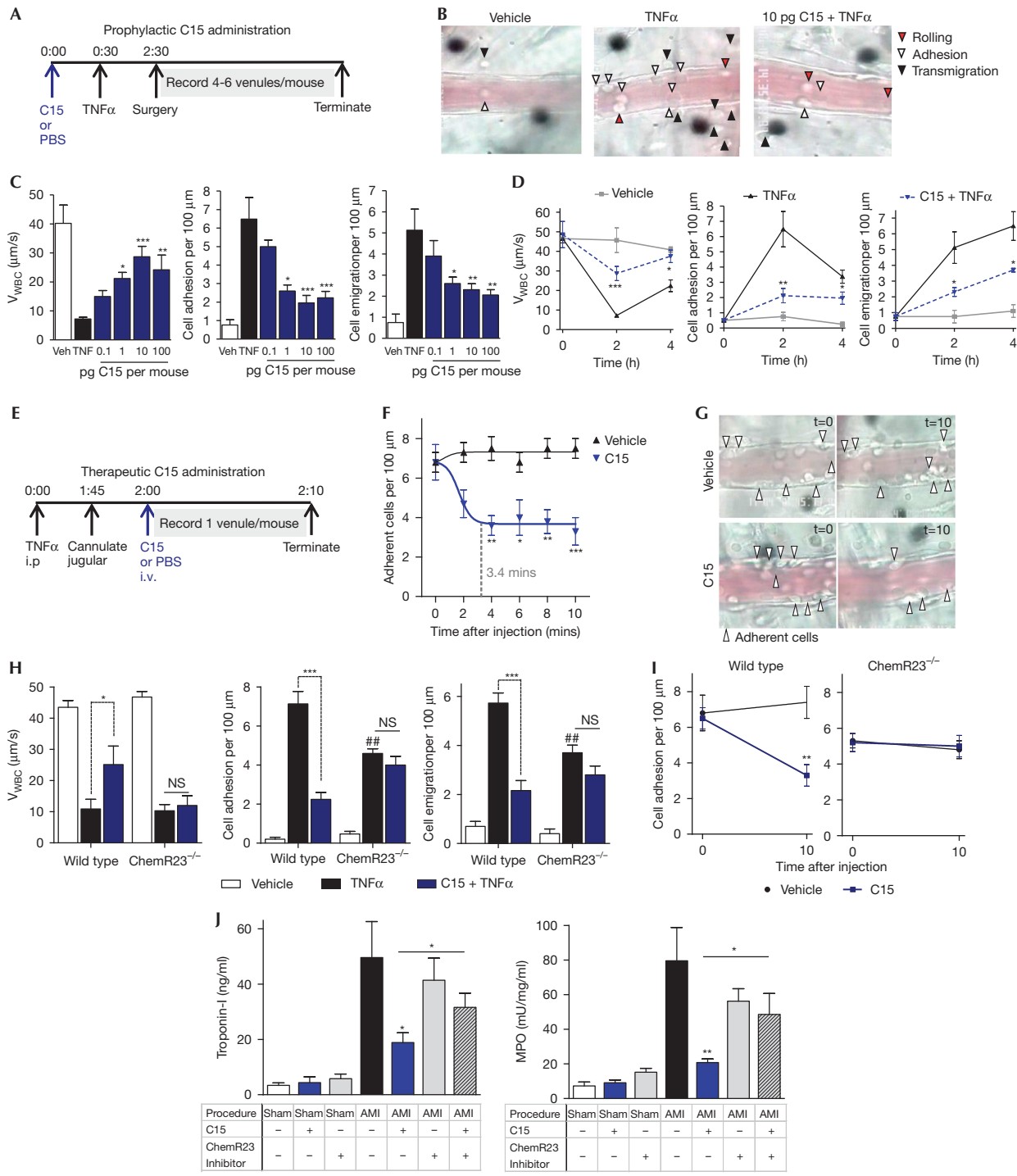

downstream β2-integrin–ICAM-1 interactions. Coupled to modulation of L-selectin shedding, these results suggest that C15 could regulate neutrophil–endothelial cell interactions under shear conditions.

### C15 inhibits human neutrophil–endothelial interactions

We used a parallel-plate flow chamber assay to study the impact of C15 on neutrophil–endothelial cell interactions under flow. C15

treatment of unstimulated neutrophils reduced cell rolling and adhesion with no significant alterations in transmigration, resulting in >50% reduction in total neutrophil–endothelial cell interactions (Fig 3A). As expected, TNFα-activated neutrophils showed reduced rolling and elevated adhesion and transmigration events versus vehicle-treated neutrophils. In this context, C15 administration decreased neutrophil adhesion and transmigration by up to 100% with a corresponding increase in rolling

◄ **Fig 4** | Chemerin15 peptide inhibits neutrophil recruitment in the inflamed vasculature and in myocardial infarction model through ChemR23 (**A**) Experimental scheme for prophylactic C15 administration. Briefly, C15 (0.1–100 pg/mouse) or vehicle (PBS) were delivered i.p., followed by i.p. TNFα (500 ng) 30 min later. Preparation of the mesentery was performed 2 h post-TNFα injection for visualization of its microcirculation. (**B**) Representative images of the 2-h time point. (**C**) Dose response to C15 (0.1–100 pg/mouse). (**D**) Time course (0–4 h) to optimal 10 pg/mouse C15 dose. (**A–D**) Extent of vascular inflammation was quantified by measuring leukocyte rolling velocities ($V_{WBC}$, μm/s), adhesion and transmigration events in 3–6 post-capillary venules per mouse. (**E**) Experimental scheme for therapeutic C15 administration. (**F**) TNFα (500 ng) was administered i.p. followed by surgery 2 h later. Once a representative vessel was located, C15 (10 pg) or vehicle was administered i.v. and adherent cells monitored over 10 min. (**G**) Representative images are shown of 0- and 10-min time points. (**H**) Effect of prophylactic C15 administration on microvascular inflammation in wild-type and ChemR23$^{-/-}$ animals. (**I**) Effect of therapeutic C15 administration on leukocyte detachment in wild-type and ChemR23$^{-/-}$ animals. (**J**) Myocardial ischaemia–reperfusion injury (30-min ischaemia: 2-h reperfusion) was performed to model myocardial infarct. ChemR23 antagonist CCX832 (2 mg/mouse) or vehicle were delivered s.c. 24 h and immediately prior to surgery. C15 (100 pg/mouse) or vehicle were delivered i.v. at the start of surgery. MPO activity and plasma Troponin-I were quantified as measures of neutrophil recruitment and heart damage. Data are expressed as means ± s.e.m. for three to eight mice per treatment group. *$P<0.05$; **$P<0.01$; ***$P<0.001$ relative to TNFα-treated or AMI animals (**C,D,H,J**) or relative to vehicle-treated animals (**I,F**) unless otherwise noted. i.p., intraperitoneal; i.v., intravenous; ns, not significant; PBS, phosphate-buffered saline; s.c., subcutaneous; TNFα, tumour necrosis factor alpha; Veh; vehicle.

events (Fig 3B). However, C15 was unable to modulate endothelial cell physiology to impact neutrophil arrest and transmigration (supplementary Fig S5 online). Collectively, we demonstrate that C15 directly targets neutrophils, particularly once in an activated state, to modulate their interactions with inflamed endothelial cells.

### C15 inhibits microvascular inflammation through ChemR23

To assess whether the effects of C15 on human and murine neutrophils *in vitro* are relevant in the context of vascular inflammation, we performed intravital microscopy of the inflamed mesenteric microcirculation. We administered TNFα, which promotes microvascular inflammation by direct activation of blood-borne neutrophils [25].

TNFα administration reduced leukocyte rolling velocities in mesenteric post-capillary venules, with concomitant increases in leukocyte adhesion (ninefold) and transmigration (sevenfold) at the optimal 2 h time point (Fig 4A–D). C15 (10 pg/mouse, intraperitoneal) administration 30 min prior to TNFα challenge counteracted the effects of this pro-inflammatory cytokine, resulting in elevated leukocyte rolling velocities (fourfold increase) and reduced neutrophil adhesion (70%) and extravasation (60%; Fig 4C; representative images shown in Fig 4B). C15 elicited these effects in a concentration-dependent manner, with maximal efficacy with as little as 10 pg or 100 pg/mouse (Fig 4C). Time-course analyses revealed that C15 accelerated the return to baseline rolling velocities while reducing neutrophil adhesion and emigration (Fig 4D).

In order to visualize a direct effect of C15 on on-going intravascular neutrophil recruitment, a situation of greater relevance to the treatment of inflammatory pathologies such as vascular injury in the clinic, we applied a real-time intravital protocol. TNFα-inflamed vessels were monitored for 10 min following intravenous administration of either saline or C15 peptide (10 pg/mouse; Fig 4E–G). In this context, C15, but not vehicle, elicited a rapid detachment of ~50% adherent neutrophils from the inflamed venular endothelium on average 3.4 min following C15 injection (Fig 4F; representative venules shown in Fig 4G).

The functional involvement of ChemR23 in these *in vivo* properties of C15 was determined using ChemR23$^{-/-}$ mice. In these animals, pre-treatment with C15 peptide was unable to modulate neutrophil rolling velocities, adhesion and transmigration in the

inflamed microcirculation (Fig 4H). The pivotal role for endogenous ChemR23 was equally evident in the real-time protocol, with an abrogation of C15-induced neutrophil detachment in ChemR23$^{-/-}$ mesenteric venules (Fig 4I). Collectively, these data demonstrate the ability of the chemerin-derived peptide, C15 to modulate neutrophil–endothelial interactions when administered prior to as well as during on-going vascular inflammation through ChemR23.

We next employed a murine model of acute myocardial infarction (AMI) to assess the relevance of the C15/ChemR23 pathway in neutrophil physiology in a clinically relevant disease model where neutrophil recruitment and β2 integrins are key pathogenic determinants [6,26,27]. As expected, AMI mouse hearts showed high myeloperoxidase activity (indicative of neutrophil infiltration) and elevated levels of Troponin-I a marker of myocardial damage used in the clinic [28]. Treatment with C15 peptide prior to AMI significantly inhibited both neutrophil myocardial infiltration and heart damage, protective effects that could be abrogated using a ChemR23 inhibitor (Fig 4J).

The data we report here for C15 provide, to our knowledge, the first description of a pro-resolving pathway that modulates neutrophil-dominated vascular inflammation in part through in-hibition of integrin activation. We thus identify the C15/ChemR23 axis as a novel therapeutic target in the treatment and/or prevention of vascular inflammation and injury. On this vein, it is tempting to propose that better understanding of how ChemR23 can be tuned towards anti-inflammatory functions could guide novel approaches for therapeutic control of aberrant neutrophil activation in disease. Further studies could address whether C15 can be used as the starting backbone for novel therapeutics, particularly for application as therapeutics in vasculitides and other forms of intravascular neutrophil activation such as reperfusion injury post-myocardial infarction or cardiac surgery.

### METHODS

Unless otherwise stated, reagents were sourced from Sigma Aldrich (Gillingham, UK).

**Chemerin peptides.** C15 (AGEDPHGYFLPGQFA) and scrambled C15 (C15-S, GLFHDQAGPPAGYEF) were synthesized by Biosynthesis (Lewisville, TX, USA), reconstituted (1 mM in PBS + 0.1% BSA) and stored at −80 °C for up to 6 months. Peptide was diluted from fresh aliquots for each experiment.

***In vitro* methods, Neutrophil isolation.** Neutrophils were isolated from human whole blood using histopaque 1,077 and 11,191, as previously described [29].

**ChemR23 staining for flow cytometry and immunofluorescence.** For determination of ChemR23 expression on human leukocytes initial experiments used two different anti-ChemR23 antibodies MAB362 (R&D Systems, Abingdon, UK; clone 84939) and clone 4C7 (gift of Prof M. Parmentier, Bruxelles, Belgium). Low ChemR23 expression was evident on resting neutrophils using both 4C7 and MAB362 and the commercially available MAB362 was used in all subsequent experiments (10 µg/ml). Full details can be found in the supplementary Methods online.. See supplementary Methods online for whole blood flow cytometry, neutrophil chemotaxis assays, phospho flow cytometry, ICAM-1-Fc binding, static ICAM-1 adhesion, static fibronectin adhesion, integrin clustering and flow chamber assays.

***In vivo* methods intravital microscopy.** C57Bl/6J mice were sourced from Charles River (Margate, UK), whereas Sv129Ev wild-type and ChemR23$^{-/-}$ mice were provided by Takeda Pharmaceuticals (Cambridge, UK). Intravital microscopy was performed as previously described, see [29] and supplementary Methods online.

**Real-time neutrophil detachment assays.** The jugular vein of anaesthetized mice was cannulated and the mesenteric vascular bed exteriorized 2 h post 500 ng intraperitoneal TNFα injection. Post-capillary venules (20–40 µm diameter) were selected and either sterile saline (100 µl) or C15 (100 µl; 0.3 ng/kg) given intravenous. Vessels were recorded for 2 min before and 10 min after intravenous injection for offline analysis. Number of adherent neutrophils were quantified every 2 min (see supplemental Methods online for description of intravital microscopy protocol).

**Murine myocardial infarction model (ischaemia–reperfusion injury).** See supplementary Methods online for description.

**Statistics.** Student's *t*-test and one-way ANOVA (with Bonferroni's *post hoc* test) were performed using GraphPad Prism 5.0 software.

ACKNOWLEDGEMENTS
This work was funded by a Sir Henry Wellcome Postdoctoral Fellowship (088967/Z/09/Z) awarded to Dr J.L.C., an Oliver Bird Studentship awarded to S.H. and a MRC PhD Studentship awarded to S.B. We thank Matt Barnes, Helen Heffron and Takeda Cambridge for generously providing ChemR23$^{-/-}$ and wild-type littermate control mice. We thank Chemocentryx (Mountain View, California) for providing ChemR23 inhibitors. We acknowledge our colleagues at the William Harvey Research Institute for guidance and advice.

  *Author contributions*: J.L.C. planned the project, designed and performed the experiments, analysed the data and wrote the manuscript; S.B. performed ischaemia–reperfusion surgery. S.H. performed and analysed the chemotaxis assays. S.M. performed the immunofluorescence experiments. V.B. performed the Ca$^{2+}$ flux experiments. M.P. contributed to the project planning and writing of the manuscript.

CONFLICT OF INTEREST
The authors declare that they have no conflict of interest.

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
