## [Review Process File · EMBO Reports]

Manuscript EMBOR-2013-37293

Chemerin15 inhibits neutrophil-mediated vascular inflammation and myocardial ischemia-reperfusion injury through ChemR23

Jenna L Cash, Stephania Bena, Sarah Headland, Simon McArthur, Vincenzo Brancaleone and Mauro Perretti

Corresponding author: Jenna L Cash, William Harvey Research Institute

Review timeline:	Submission date:	19 March 2013
	Editorial Decision:	09 April 2013
	Correspondence:	17 April 2013
	Correspondence:	30 April 2013
	Revision received:	30 July 2013
	Accepted:	02 August 2013

Editor: Nonia Pariente

Transaction Report:

1st Editorial Decision

09 April 2013

Thank you for your submission to EMBO reports. We have now received reports from the three referees that were asked to evaluate your study, which can be found at the end of this email. As you will see, although all the referees find the topic of interest, they all consider the study preliminary for publication here at this stage. Referees 1 and 2 raise a substantial number of serious concerns about the conclusiveness of the results, and request a number of technical improvements of the data, and referee 3 considers that further experiments are needed to provide proof of the physiological relevance of your study. In addition, referee 1 raises concerns regarding the overall presentation of the work.

From the analysis of these comments, it is clear that publication of your manuscript in our journal cannot be considered at this stage. On the other hand, given the potential interest of your study, we would be willing to give you the opportunity to address the reviewers concerns and with the understanding that the referee concerns must be adequately addressed and that acceptance of the manuscript would entail a second round of review. It is EMBO reports policy to undergo one round of revision only and thus, acceptance of your study would depend on the outcome of the next, final round of peer-review.

I appreciate that addressing the referees comments would involve a lot of additional experimental work of unclear outcome, in particular addressing points 4-7 of referee 1, the concerns of referee 2 regarding disambiguation of beta2-integrins, and effects on integrin affinity and/or avidity, as well as providing evidence of the importance of the C15-Chem23 axis in the context of disease. Providing a mechanism for how C15 inhibits B2-integrin activation or the potential relationship between the modulation of selectins and integrins may be out of the scope of a short report and could be discussed in the text. However, all other issues would have to be experimentally addressed. Please note that although referee 1 contemplates the possibility of toning down the part of interstitial migration in his/her point 7, we would require that an alternative explanation be experimentally provided.

I am unsure if you will be able to return a revised manuscript within our 3 months deadline and could potentially somewhat extend our deadline for revision [should you feel time is the only limitation to a successful revision of the paper]. However, I would also understand your decision if you chose to rather seek rapid publication elsewhere at this stage.

If you decide to revise your study, revised manuscripts must be submitted within three months of a request for revision unless previously discussed with the editor; they will otherwise be treated as new submissions. Revised manuscript length must be a maximum of 28,500 characters (including spaces). When submitting your revised manuscript, please also include editable TIFF or EPS-formatted figure files, a separate PDF file of any Supplementary information (in its final format) and a letter detailing your responses to the referees.

We also welcome the submission of cover suggestions or motifs that might be used by our Graphics Illustrator in designing a cover.

I look forward to seeing a revised form of your manuscript when it is ready. In the meantime, do not hesitate to get in touch with me if I can be of any assistance.

REFeree REPORTS:

Referee #1:

Cash et al, Chemerin peptide inhibits neutrophil activation and vascular inflammation by regulating beta2 integrin activation

In the present paper Cash et al propose that chemerin15 (C15), an anti-inflammatory peptide derived from chemerin which binds to the ChemR23 receptor, restricts excessive neutrophil-dependent inflammation by limiting beta2 integrin-dependent neutrophil trafficking. The authors demonstrate dynamic neutrophil ChemR23 expression on both human and mouse neutrophils, taking advantage of a ChemR23-deficient mouse model. Both selectin shedding and integrin inside-out signalling are downregulated by C15 peptide treatment of neutrophils. This leads to reduced integrin-dependent outside-in signalling (adhesion and spreading, chemotaxis, Erk phosphorylation) in vitro. In-line with this, C15 peptide treatment leads to reduced neutrophil adhesion and transendothelial migration in flow assays and in vivo as observed by intravital imaging. This is an interesting body of work. However the work is not suitable to be published in its current shape. The present manuscript is very hard to read. This interferes with understanding the data shown in some of the figures even on repeated reading. Introduction, some of the methodology, figure legends, and discussion should all be expanded to allow the wide readership of EMBO Reports to fully appreciate this work. For instance, the use of affinity status specific antibodies to allow the analysis of the ligand binding affinity status of integrins on human neutrophils might be guessed by those connected to, but not by those new to the field.

Specific points

- (1) The authors show that ChemR23 expression is dynamically upregulated in neutrophils on activation with pro- but not with anti-inflammatory mediators. Why does the peptide downregulate inflammation only in pro-inflammatory contexts?
- (2) The authors state that ChemR23 is punctate and co-localises with secretory and specific granules in resting neutrophils. This is backed up by some micrographs. Method, text and figure legend need to be increased to allow the reader to appreciate the experiments carried out and what they show. What are these images, how were they generated, how were they analysed? A graph should indicate which percentage of cells behaved in the same way as the representative examples shown.
- (3) It is intriguing that C15 affects both selectins and integrins. Are the modulations of selectins and integrins separate events or are they causally linked? Have any potential mechanisms been explored (e.g. Syk signalling)? If no mechanistic insight can be presented, these possibilities should at least be discussed.
- (4) The fact that C15 affects not just Mac-1 but also other integrins' inside-out signalling should be expanded, the experiment should be repeated with murine neutrophils taking advantage of the ChemR23^{-/-} cells and the data included in a main figure.
- (5) Fig 2L shows reduced Erk phosphorylation on treatment with C15 peptide and TNF α stimulation. No further explanations to this experiment are given, and it is not clear whether adhesion-dependent Erk phosphorylation was being monitored. This should be clarified. To demonstrate that Erk phosphorylation is indeed integrin dependent in the context of the assay, appropriate blocking antibodies could be employed.
- (6) There are numerous reports demonstrating that different chemotaxis assays vary in terms of the integrin-dependency of the chemotaxis they measure. Carbo et al (J Leukoc Biol 88, 313) demonstrated recently that neutrophil chemotaxis in transwells is integrin-independent. The authors should therefore make use of a different chemotaxis assay in Fig 2K (and Fig S2) to demonstrate a beta2 integrin-dependent chemotaxis defect on treatment of cells with C15 peptide.
- (7) Similarly, interstitial leukocyte migration has recently been shown genetically to be entirely integrin-independent (Lammermann et al, Nature 453, 51-55). Is it possible that actomyosin contractility or another cellular attribute important for migration in three dimensional matrices is affected in C15 peptide treated neutrophils? Alternative mechanisms could be addressed experimentally to identify the mechanism underlying the migration defects the authors observed. In the absence of a mechanism, the current strong statement should be replaced by a more moderate one and the data discussed in the lights of the relevant literature.
- (8) Abbreviations need to be defined and figure legends re-written to contain enough details (as those for the supplemental figures do already).

Referee #2:

In this study, Cash et al. investigate the effects of chemerin15 on neutrophil activation and recruitment. By using different *in vitro* and *in vivo* assays, they demonstrate that C15 has anti-inflammatory properties. Binding of C15 to its receptor reduces neutrophil adhesion *in vitro* and *in vivo* by modulating β 2-integrin activation. The study has several limitations and the conclusions are not always supported by the presented data.

It has been shown before that chemerin and its receptor ChemR23 can suppress neutrophil recruitment into inflamed tissue by altering the expression of pro-inflammatory cytokines. In this report, the authors show an additional mechanism how C15 can modulate neutrophil recruitment, but the presented data are not conclusive. In this study, the authors show that the stimulation of PMN with C15 increases intracellular Ca²⁺-levels. However, an increase of Ca²⁺-levels, as seen in neutrophils after stimulation with CXCL1, is associated with the activation of neutrophils leading to β 2-integrin activation. Furthermore, L-selectin shedding is also seen in stimulated PMN. The authors should investigate the signaling pathway triggered by C15 and provide a mechanism how C15 inhibits β 2-integrin activation.

As the authors focus on the different steps of the recruitment cascade, they should also distinguish between the two beta2-integrins (LFA-1 and Mac-1) expressed on neutrophils. After activation, both integrins can bind ICAM-1, but the authors did not investigate which integrin is inhibited by C15. Furthermore, the authors should investigate whether C15 modulates integrin affinity (human reporter antibodies) and/or avidity (clustering).

Referee #3:

The study by Cash et al. provides novel evidence for the importance of the C15-ChemR23 in beta2-integrin dependent neutrophil recruitment. Rationale and data display is clear and the data are convincing. To further improve their study the authors should provide evidence for the importance of the C15-ChemR23 axis in disease models where neutrophil adhesion via beta2-integrins was shown to be important. Such could be done in models of atherosclerosis (e.g. IVM of carotid artery) or acute lung injury (IVM of lung microvasculature or 3-compartment model).

Correspondence - authors

17 April 2013

Re: EMBOR-2013-37293V1. Chemerin15 peptide inhibits neutrophil activation and vascular inflammation by regulating β 2 integrin activation

We are grateful to yourself and the reviewers for their careful reading of the manuscript and the helpful comments provided. In particular, we thank you for identifying the novelty(ies) contained in this study and summarizing what we should do to raise its impact. Alongside your suggestion, and in agreement with the other Authors, we would like to take the opportunity to address the comments and revise the manuscript. We propose to address the comments in the following way:

1. We will rewrite the manuscript as suggested by Reviewer 1 to make it easier to read.
2. We will attempt to provide some mechanistic insight into the ability of C15 to inhibit integrin activation by assessing Syk signalling and including more information on ERK phosphorylation (Reviewer 1, points 3&5).
3. We will assess the impact of C15 on the ability of murine neutrophils (wildtype & ChemR23^{-/-}) to bind β 1 integrin ligand (Reviewer 1, point 4).
4. We will perform new chemotaxis assays to determine the effect of C15 on real time neutrophil chemotaxis on the β 2 integrin ligand ICAM-1 (Reviewer 1, point 6).
5. Flow cytometry experiments will be repeated with antibodies that specifically detect the active conformation of LFA-1 as well as total LFA-1 (CD11a; Reviewer 2).
6. We have already investigated the effect of C15 on integrin affinity using antibodies that detect the high affinity (extended) conformation of CD18 and CD11b. We will add to this by assessing the effect of C15 on β 2 integrin clustering on neutrophils (Reviewer 2).
7. To further improve the study, as requested by Reviewer 3, we will use a murine model of myocardial ischemia-reperfusion injury in which the importance of neutrophil adhesion and β 2 integrins is established, to study the role of the C15/ChemR23 axis.
8. With respect to neutrophil interstitial migration the role of integrins is not clear. Lammerman et al (2008) have demonstrated integrin-independent interstitial migration for Dendritic cells, however other studies on neutrophils (eg. Werr, J Exp Med, 1998) have illustrated a contributory role for integrins in interstitial migration. In light of the current lack of consensus on the role of integrins in interstitial migration and to simplify the message of the manuscript, we propose removal of interstitial migration data. We feel that in this manner we will improve the clarity of the manuscript by focusing on data and assays where the role of integrins is more 'concrete'.

Additionally, we will of course, address, the other minor comments made by the Reviewers.

We will do our utmost to adhere to the 3 month resubmission limit. However, if the outlined experiments appear to be taking us close to this limit we would like, as you suggested, the opportunity to request a small extension to this deadline. Once again, we would like to thank you for giving us this opportunity.

Thank you for your email. I am sorry for the misunderstanding, as there was no explicit question associated with your letter, I did not think you expected an answer.

The revision plans sound reasonable indeed and we could accept your decision to refocus the work and omit the interstitial migration data. As you mention, however, the final decision will depend on the outcome of the peer-review process and, thus, on the results you obtain in trying to address the referee concerns. These should be sufficiently insightful (with regards to understanding of C15-induced effects) and demonstrate an important role of the C15-ChemR23 axis in beta2-integrin dependent neutrophil recruitment in a disease-relevant setting. In this regard, it would be ideal to use one of the two models suggested by reviewer 3, if at all possible, as they are of broad general interest. However, if this is not feasible, I would think that reviewer 3 would probably be satisfied by the inclusion of data on myocardial ischemia-reperfusion injury.

I hope my response is of help in preparing your revision for EMBO reports. As a matter of policy, we only invite revision of those studies that stand a good chance of acceptance after review, so if revision is conscientiously and thoroughly performed, I think this would be the case here.

VERSION 2 EMBOR-2013-37293

Thank you to each of the Referees for their careful reviewing of our manuscript. We feel that the extensively revised manuscript has been markedly improved by taking on board and addressing the concerns noted on our original submission.

For each referee we have copy/pasted the concerns and questions below in **bold** and numbered each point. Our replies are in standard text.

General points:

Fig 2 had 12 panels in the original submission and to address referees comments the results from 4 new assays need to be inserted into this figure. These are integrin-dependent migration assay, signalling assays, $\beta 1$ integrin adhesion assay and integrin clustering. In order to accommodate most of these new data we have restructured the figure removing some of the original data (Fig.2A, D and E) to supplementary data. Our feeling is that if these data had remained in the main figure then it would become overwhelming to readers, thus we have tried to change it to make it easily accessible and understandable to the readership.

Referee #1:

1). The present manuscript is very hard to read. This interferes with understanding the data shown in some of the figures even on repeated reading. Introduction, some of the methodology, figure legends, and discussion should all be expanded to allow the wide readership of EMBO Reports to fully appreciate this work. For instance, the use of affinity status specific antibodies to allow the analysis of the ligand binding affinity status of integrins on human neutrophils might be guessed by those connected to, but not by those new to the field.

We thank Referee 1 for their pertinent comments. With respect to the manuscript being hard to read we apologise for this, obviously it wasn't our intention. We hope we have addressed this to your satisfaction in the revised manuscript. We have also restructured Figure 2 in particular to improve flow.

All changes are marked in red, we made many modifications to address this point so have not specifically referred to each point in this document.

2). The authors show that ChemR23 expression is dynamically upregulated in neutrophils on activation with pro- but not with anti-inflammatory mediators. Why does the peptide downregulate inflammation only in pro-inflammatory contexts?

The peptide does affect the behaviour of neutrophils both in their resting state (Fig.2A L-selectin and PSGL-1 expression and Fig.3A inhibition of neutrophil rolling under flow) and once activated, however, the effect of C15 is more pronounced on activated neutrophils, probably due to upregulation of the receptor ChemR23.

With respect to the upregulation of ChemR23 on neutrophils with pro- but not anti-inflammatory mediators (similar to FPR2; annexin receptor (1)) our working hypothesis is that in order to contain the magnitude of an anti-inflammatory response and thus in part prepare for resolution, anti-inflammatory receptors are upregulated before resolution begins.

(3) It is intriguing that C15 affects both selectins and integrins. Are the modulations of selectins and integrins separate events or are they causally linked? Have any potential mechanisms been explored (e.g. Syk signalling)? If no mechanistic insight can be presented, these possibilities should at least be discussed.

It is known that L-selectin crosslinking can promote neutrophil $\beta 2$ integrin activation. Therefore L-selectin shedding may reduce crosslinking and $\beta 2$ integrin activation to limit inflammation (2). We have not performed experiments to determine whether modulation of selectins and integrins by C15 are separate events or not as we concur with the Editor that this may be outside of the scope of a brief report.

However, we have performed several new experiments to address the mechanism. Please see point 5 below, response to Reviewer 2 point 2 and Fig.2K.

(4) The fact that C15 affects not just Mac-1 but also other integrins' inside-out signalling should be expanded, the experiment should be repeated with murine neutrophils taking advantage of the ChemR23^{-/-} cells and the data included in a main figure.

Thank you for your suggestion. We have used wildtype and ChemR23^{-/-} neutrophils and assessed static adhesion to the $\beta 1$ integrin ligand fibronectin. We observed a significant inhibition of adhesion in wildtype but not ChemR23^{-/-} neutrophils (Fig.S3B). $\beta 3$ integrins are not known to be expressed on neutrophils.

Figure 2 is a difficult figure to follow partly as we have a mesh of mouse and human neutrophil work, we monitored expression of several molecules and used ChemR23 inhibitors and ChemR23^{-/-} mice. We're worried that by including $\beta 1$ integrin studies in some but not all assays used to assess effects on $\beta 2$ integrins that the reader will be confused. We felt the size of the figure was so large already that we could not repeat all the assays for $\beta 1$ integrins. We have therefore kept all $\beta 1$ integrin data in supplementary figures 3 (neutrophils) and 4 (monocytes).

We hope you agree that to address your very valid point (numbered 1 above) of making the manuscript and figures easier to follow that this is the best route. However, we are of course open to putting the data of supplementary Fig.3 in Figure 2 if necessary.

(5) Fig 2L shows reduced Erk phosphorylation on treatment with C15 peptide and TNFalpha stimulation. No further explanations to this experiment are given and it is not clear whether adhesion-dependent Erk phosphorylation was being monitored. This should be clarified. To demonstrate that Erk phosphorylation is indeed integrin dependent in the context of the assay, appropriate blocking antibodies could be employed.

We accept that our experimental plan was not clear and we have now provided full details in the Supplementary Methods.

Further detail of the mechanism for C15-mediated inhibition of integrin activation is suggested by Referee 2 in point 2 as well as by yourself. We have therefore extended our studies looking not only at ERK phosphorylation, but also Syk and Src using an intracellular flow cytometry approach which is higher throughput and quicker than western blotting.

Syk, ERK and Src kinases are known to mediate inside-out (and outside-in) signalling leading to $\beta 2$ integrin activation and clustering (3-7). pSyk (Y525/526) and pERK (Y202/204) represent activated enzymes whilst Y527 of Src is constitutively phosphorylated to maintain the enzyme in an inactive state, thus dephosphorylation of Y527 is associated with Src activation (8). Neutrophil stimulation with TNF α activated all three enzymes by triggering Syk and ERK phosphorylation and Src de-phosphorylation. We found that neutrophil treatment with 10 pM C15 prevented TNF α -induced activation of Syk (69 %), ERK (80 %) and Src (61 %; Fig.2K).

With respect to your question regarding integrin dependency of the signalling events, this is a very important point, thank you for raising it. Integrin signalling is bidirectional, thus inside-out signalling triggers integrin activation whereas binding of activated integrin to substrate triggers outside-in signalling which modulates various aspects of the cells behaviour including survival (6, 9). To avoid looking at the latter we performed signalling experiments in suspension in the absence of soluble integrin ligand.

In order for C15 to affect integrin activation induced by eg. TNF α it is inhibiting aspects of inside-out signalling (inhibiting the signals downstream of TNF receptor but upstream of integrin-mediated adhesion to integrin ligand) the consequence of which is inhibition of integrin-ligand binding and neutrophil adhesion (6, 9). Thus we have not looked at adhesion-dependent Erk phosphorylation as we are inhibiting adhesion and the signalling required to achieve this affect occurs upstream of the adhesion event.

(6) There are numerous reports demonstrating that different chemotaxis assays vary in terms of the integrin-dependency of the chemotaxis they measure. Carbo et al (J Leukoc Biol 88, 313) demonstrated recently that neutrophil chemotaxis in transwells is integrin-independent. The authors should therefore make use of a different chemotaxis assay to demonstrate a beta2 integrin-dependent chemotaxis defect on treatment of cells with C15 peptide.

Thank you for raising this important point. To address it we have removed the original chemotaxis data using neuroprobe chemotaxis plates and have setup a new assay using ICAM-1 coated IBIDI μ -slide chemotaxis chambers. In this system ICAM-1 -dependent neutrophil chemotaxis towards fMLF is monitored by live cell tracking. We've found this system to be an impressive and much more informative way of assessing chemotaxis *in vitro*.

With these new experiments we found that C15 treatment significantly impaired ICAM-1-dependent neutrophil chemotaxis towards fMLF as shown by plots of the trajectory paths in a new Fig. 2I. This was quantified by measuring the centre of mass (spatial averaged point of all cell endpoints) as an indicator of cell directionality and velocity (Fig.2J). We therefore demonstrate that C15 inhibits $\beta 2$ integrin-dependent neutrophil chemotaxis.

(7) Similarly, interstitial leukocyte migration has recently been shown genetically to be entirely integrin-independent (Lammermann et al, Nature 453, 51-55). Is it possible that actomyosin contractility or another cellular attribute important for migration in three dimensional matrices is affected in C15 peptide treated neutrophils? Alternative mechanisms could be addressed experimentally to identify the mechanism underlying the migration defects the authors observed. In the absence of a mechanism, the current strong statement should be replaced by a more moderate one and the data discussed in the lights of the relevant literature.

Thank you for highlighting this point. With respect to neutrophil interstitial migration the role of integrins is not clear. Lammermann et al (10) have demonstrated integrin-independent interstitial migration for Dendritic cells, however other studies on neutrophils have illustrated a role for integrins in interstitial migration (11-13). After discussions with the Editor, due to the current lack of

consensus on the role of integrins in interstitial migration and to simplify the message of the manuscript, we have decided to remove the interstitial migration data. We feel that in this manner we have improved the clarity of the manuscript by focusing on data and assays where the role of integrins is more 'concrete'.

8) The authors state that ChemR23 is punctate and co-localises with secretory and specific granules in resting neutrophils. This is backed up by some micrographs. Method, text and figure legend need to be increased to allow the reader to appreciate the experiments carried out and what they show. What are these images, how were they generated, how were they analysed? A graph should indicate which percentage of cells behaved in the same way as the representative examples shown.

Thank you for your suggestion. We have improved the method, text and figure legend to make it clearer what experiments we have done and their results. We have not included a graph to indicate the percentage of cells that behaved in the same way as the examples as virtually every cell behaved in this way so a graph would convey little meaningful information for the reader. Instead we have stated this in the Supplementary Methods.

(9) Abbreviations need to be defined and figure legends re-written to contain enough details (as those for the supplemental figures do already).

We have re-written the figure legends and modified our abbreviations to improve clarity.

Referee #2:

1A). In this study, the authors show that the stimulation of PMN with C15 increases intracellular Ca²⁺-levels. However, an increase of Ca²⁺-levels, as seen in neutrophils after stimulation with CXCL1, is associated with the activation of neutrophils leading to β2-integrin activation.

Thank you for raising this point. Calcium flux responses in neutrophils can be induced by both pro- and anti-inflammatory mediators as well as pro-resolving molecules. An increase in cellular calcium levels indicates release of calcium from intracellular stores such as the endoplasmic reticulum or influx from the extracellular environment through ion channels which is triggered by activation of a receptor. In the case of GPCRs, some are coupled to Gq and Gi proteins (including ChemR23 (14) and CXCL1 receptor, CXCR2 (15)) which stimulate intracellular calcium flux. Thus a calcium flux response is not indicative of cell or β2 integrin activation but due to the specific G protein coupled to the ligands receptor (16).

Other anti-inflammatory mediators that induce calcium flux responses in neutrophils but do not activate neutrophils, include Annexin A1 and Ac2-26 (17).

We have made our intentions with this assay clearer in the text (page 6, paragraph 2). These are that since the expression of ChemR23 on neutrophils is an entirely novel finding we attempted to solidify our data by demonstrating that C15 can induce signalling on neutrophils via ChemR23 thus illustrating that the *functional* receptor is expressed on neutrophils.

1B). Furthermore, L-selectin shedding is also seen in stimulated PMN.

Indeed, L-selectin shedding is observed on stimulated (activated) neutrophils in response to pro-inflammatory mediators. However, L-selectin shedding is also seen in response to anti-inflammatory mediators without concomitant cell activation. Examples include non-steroidal anti-inflammatory drugs, dexamethasone and annexin A1 (18-20).

Several studies have demonstrated *in vitro* and *in vivo* that L-selectin shedding results in higher neutrophil rolling velocities (21, 22). L-selectin crosslinking is also thought to participate in

leukocyte activation by enhancing $\beta 2$ integrin activation; thus L-selectin shedding limits this activation and may therefore limit inflammation (2).

We have now included further background to L-selectin in the text to ensure the significance of its shedding is readily understandable (page 7, paragraph 1).

2). The authors should investigate the signaling pathway triggered by C15 and provide a mechanism how C15 inhibits $\beta 2$ -integrin activation.

We have tried to investigate the molecular pathways involved in C15's inhibitory effects on integrin activation, however this is a difficult point to address, partly as the pathways have only partially been elucidated and also due to technical difficulties.

Syk, Erk and Src kinases are known to mediate inside-out (and outside-in) signalling leading to $\beta 2$ integrin activation and clustering (3-7). We used an intracellular flow cytometry approach to assess the phosphorylation status of these enzymes. pSyk (Y525/526) and pERK (Y202/204) represent activated enzymes whilst Y527 of Src is constitutively phosphorylated to maintain the enzyme in an inactive state, thus dephosphorylation of Y527 results is associated with Src activation. Neutrophil stimulation with TNF α activated all three enzymes by triggering Syk and ERK phosphorylation and Src de-phosphorylation. We found that neutrophil treatment with 10 pM C15 prevented TNF α -induced activation of Syk (69 %), ERK (80 %) and Src (61 %; Fig.2)].

Unsuccessful approaches - We tried to look at talin and kindlin-3 localisation to the membrane and cytosol compartments as these proteins can bind to the cytoplasmic portion of integrins to elicit activation and thus, in theory, be retained in the membrane fraction. However technical difficulties rendered this approach unsuccessful. Further, we tried to assess Rap1 activation. Rap1-GTP plays a role in activating talin to enable its binding to integrin. Although we were able to detect Rap1 activation in activated platelets, we were unable to observe any signal in neutrophil extracts using multiple time points, activators and lysis buffers etc. We have shown the Rap1 data for the reviewers benefit below (Fig.1).

Figure for Reviewers 1: Western blot showing active Rap1 (GTP-Rap1) and total Rap1 levels in human neutrophil and platelet lysates.

3. As the authors focus on the different steps of the recruitment cascade, they should also distinguish between the two beta2-integrins (LFA-1 and Mac-1) expressed on neutrophils. After activation, both integrins can bind ICAM-1, but the authors did not investigate which integrin is inhibited by C15.

Thank you for raising this point. We have repeated flow cytometry experiments using MEM-83 antibody which binds to the activation epitope in the I-domain of αL (LFA-1 alpha subunit) and thus detects the high affinity form of CD11a.

Neutrophil pre-treatment with C15 prior to stimulation with TNF α led to significant inhibition of CD11a activation (Fig.2B) and clustering (Fig.2C, see point 4 below).

4. Furthermore, the authors should investigate whether C15 modulates integrin affinity (human reporter antibodies) and/or avidity (clustering).

In our original submission we demonstrate that C15 modulates integrin affinity antibodies mAb24 and CBRM1/4 which detect activated CD18 and CD11b respectively. We have now also assessed CD11a affinity modulation as requested above (see Fig.2B).

As requested, we have also assessed clustering of CD11a and CD11b by immunofluorescence and demonstrate that C15 potently inhibits clustering of both integrins, but with a greater effect on CD11a. This data is included in a new Fig.2C.

Referee #3:

The study by Cash et al. provides novel evidence for the importance of the C15-ChemR23 in beta2-integrin dependent neutrophil recruitment. Rationale and data display is clear and the data are convincing.

1. To further improve their study the authors should provide evidence for the importance of the C15-ChemR23 axis in disease models where neutrophil adhesion via beta2-integrins was shown to be important. Such could be done in models of atherosclerosis (e.g. IVM of carotid artery) or acute lung injury (IVM of lung microvasculature or 3-compartment model).

Thank you for noting the novelty of our data, clear display and convincing nature. We took your valid comment on board and have now performed new experiments to assess the importance of C15-ChemR23 axis in a disease model.

We chose a murine acute myocardial infarction model (AMI, myocardial ischemia-reperfusion injury) where the role of neutrophil recruitment and β 2-integrins is established (23-27).

We chose not to use the acute lung injury model as our reading of the literature revealed that neutrophil recruitment in this model can be either β 2-integrin dependent OR independent whereas this ambiguity is, to our knowledge, not evident for the AMI injury model (28). IVM of the carotid artery (atherosclerosis) is an excellent suggestion, however, due to this model being of the chronic variety and requiring additional time to set up new collaborations, we were sure that we would not be able to complete the experiments in a timely fashion, as required by the Journal.

We were unable to use the ChemR23^{-/-} mice in these studies as the time required to breed sufficient animals would not have allowed us to revise the manuscript within the stipulated revision deadline. We have instead used a ChemR23 antagonist.

In our new experiments, treatment with C15 peptide prior to AMI significantly inhibited neutrophil myocardial infiltration (myeloperoxidase activity) with concomitant reductions in plasma troponin-I, a clinical marker of heart damage (29)). These protective effects could be abrogated using a ChemR23 inhibitor (Fig.4J). We have therefore demonstrated that C15, through ChemR23, can exert protective effects in acute myocardial infarction.

References

1. Nadkarni S, Cooper D, Brancaleone V, Bena S, Perretti M. Activation of the annexin A1 pathway underlies the protective effects exerted by estrogen in polymorphonuclear leukocytes. *Arteriosclerosis, thrombosis, and vascular biology*. 2011;31(11):2749-59. Epub 2011/08/13.
2. Hafezi-Moghadam A, Thomas KL, Prorock AJ, Huo Y, Ley K. L-selectin shedding regulates leukocyte recruitment. *The Journal of experimental medicine*. 2001;193(7):863-72. Epub 2001/04/03.

3. Bouaouina M, Blouin E, Halbwachs-Mecarelli L, Lesavre P, Rieu P. TNF-induced beta2 integrin activation involves Src kinases and a redox-regulated activation of p38 MAPK. *J Immunol.* 2004;173(2):1313-20. Epub 2004/07/09.
4. Luttrell DK, Luttrell LM. Not so strange bedfellows: G-protein-coupled receptors and Src family kinases. *Oncogene.* 2004;23(48):7969-78. Epub 2004/10/19.
5. Piccardoni P, Sideri R, Manarini S, Piccoli A, Martelli N, de Gaetano G, et al. Platelet/polymorphonuclear leukocyte adhesion: a new role for SRC kinases in Mac-1 adhesive function triggered by P-selectin. *Blood.* 2001;98(1):108-16. Epub 2001/06/22.
6. Harburger DS, Calderwood DA. Integrin signalling at a glance. *Journal of cell science.* 2009;122(Pt 2):159-63. Epub 2009/01/02.
7. Li Z, Zhang G, Feil R, Han J, Du X. Sequential activation of p38 and ERK pathways by cGMP-dependent protein kinase leading to activation of the platelet integrin alphaIIb beta3. *Blood.* 2006;107(3):965-72. Epub 2005/10/08.
8. Arias-Romero LE, Saha S, Villamar-Cruz O, Yip SC, Ethier SP, Zhang ZY, et al. Activation of Src by protein tyrosine phosphatase 1B is required for ErbB2 transformation of human breast epithelial cells. *Cancer research.* 2009;69(11):4582-8. Epub 2009/05/14.
9. Plow EF. Inside-out, outside-in: what's the difference? *Blood.* 2007(109):3128-9.
10. Lammermann T, Bader BL, Monkley SJ, Worbs T, Wedlich-Soldner R, Hirsch K, et al. Rapid leukocyte migration by integrin-independent flowing and squeezing. *Nature.* 2008;453(7191):51-5. Epub 2008/05/03.
11. Werr J, Xie X, Hedqvist P, Ruoslahti E, Lindbom L. beta1 integrins are critically involved in neutrophil locomotion in extravascular tissue *In vivo.* *The Journal of experimental medicine.* 1998;187(12):2091-6. Epub 1998/06/24.
12. Tong H, Zhao B, Shi H, Ba X, Wang X, Jiang Y, et al. c-Abl tyrosine kinase plays a critical role in beta2 integrin-dependent neutrophil migration by regulating Vav1 activity. *Journal of leukocyte biology.* 2013;93(4):611-22. Epub 2013/01/18.
13. Kumar S, Xu J, Perkins C, Guo F, Snapper S, Finkelman FD, et al. Cdc42 regulates neutrophil migration via crosstalk between WASp, CD11b, and microtubules. *Blood.* 2012;120(17):3563-74. Epub 2012/08/31.
14. Wittamer V, Franssen JD, Vulcano M, Mirjolet JF, Le Poul E, Migeotte I, et al. Specific recruitment of antigen-presenting cells by chemerin, a novel processed ligand from human inflammatory fluids. *The Journal of experimental medicine.* 2003;198(7):977-85. Epub 2003/10/08.
15. Raghuwanshi SK, Su Y, Singh V, Haynes K, Richmond A, Richardson RM. The chemokine receptors CXCR1 and CXCR2 couple to distinct G protein-coupled receptor kinases to mediate and regulate leukocyte functions. *J Immunol.* 2012;189(6):2824-32. Epub 2012/08/08.
16. Arkin MC, PR.; Emkey, R et al. . FLIPR™ Assays for GPCR and Ion Channel Targets. In: Sittampalam GS, Gal-Edd N, Arkin M, et al, editors *Assay Guidance Manual [Internet] Bethesda (MD): Eli Lilly & Company and the National Center for Advancing Translational Sciences; 2004-* Available from: <http://www.ncbi.nlm.nih.gov/books/NBK92012/>. 2012.
17. Bena S, Brancaleone V, Wang JM, Perretti M, Flower RJ. Annexin A1 interaction with the FPR2/ALX receptor: identification of distinct domains and downstream associated signaling. *The Journal of biological chemistry.* 2012;287(29):24690-7. Epub 2012/05/23.
18. Strausbaugh HJ, Rosen SD. A potential role for annexin 1 as a physiologic mediator of glucocorticoid-induced L-selectin shedding from myeloid cells. *J Immunol.* 2001;166(10):6294-300. Epub 2001/05/09.
19. Gomez-Gaviro MV, Gonzalez-Alvaro I, Dominguez-Jimenez C, Peschon J, Black RA, Sanchez-Madrid F, et al. Structure-function relationship and role of tumor necrosis factor-alpha-converting enzyme in the down-regulation of L-selectin by non-steroidal anti-inflammatory drugs. *The Journal of biological chemistry.* 2002;277(41):38212-21. Epub 2002/07/31.
20. de Coupade C, Solito E, Levine JD. Dexamethasone enhances interaction of endogenous annexin 1 with L-selectin and triggers shedding of L-selectin in the monocytic cell line U-937. *British journal of pharmacology.* 2003;140(1):133-45. Epub 2003/09/12.
21. Hafezi-Moghadam A, Ley K. Relevance of L-selectin shedding for leukocyte rolling *in vivo.* *The Journal of experimental medicine.* 1999;189(6):939-48. Epub 1999/03/17.

22. Walcheck B, Kahn J, Fisher JM, Wang BB, Fisk RS, Payan DG, et al. Neutrophil rolling altered by inhibition of L-selectin shedding in vitro. *Nature*. 1996;380(6576):720-3. Epub 1996/04/25.
23. Simpson PJ, Todd RF, 3rd, Mickelson JK, Fantone JC, Gallagher KP, Lee KA, et al. Sustained limitation of myocardial reperfusion injury by a monoclonal antibody that alters leukocyte function. *Circulation*. 1990;81(1):226-37. Epub 1990/01/01.
24. Jordan JE, Zhao ZQ, Vinten-Johansen J. The role of neutrophils in myocardial ischemia-reperfusion injury. *Cardiovascular research*. 1999;43(4):860-78. Epub 2000/01/01.
25. Litt MR, Jeremy RW, Weisman HF, Winkelstein JA, Becker LC. Neutrophil depletion limited to reperfusion reduces myocardial infarct size after 90 minutes of ischemia. Evidence for neutrophil-mediated reperfusion injury. *Circulation*. 1989;80(6):1816-27. Epub 1989/12/01.
26. Palazzo AJ, Jones SP, Girod WG, Anderson DC, Granger DN, Lefer DJ. Myocardial ischemia-reperfusion injury in CD18- and ICAM-1-deficient mice. *The American journal of physiology*. 1998;275(6 Pt 2):H2300-7. Epub 1998/12/09.
27. Frangogiannis NG, Smith CW, Entman ML. The inflammatory response in myocardial infarction. *Cardiovascular research*. 2002;53(1):31-47. Epub 2001/12/18.
28. Grommes J, Soehnlein O. Contribution of neutrophils to acute lung injury. *Mol Med*. 2011;17(3-4):293-307. Epub 2010/11/04.
29. Keller T, Zeller T, Peetz D, Tzikas S, Roth A, Czyn E, et al. Sensitive troponin I assay in early diagnosis of acute myocardial infarction. *The New England journal of medicine*. 2009;361(9):868-77. Epub 2009/08/28.

2nd Editorial Decision

02 August 2013

Thank you for the submission of your revised manuscript to EMBO reports. It has now been seen by referees 2 and 3 (referee was unavailable), who support publication and have no further comments. I am thus very pleased to accept your manuscript for publication in the next available issue of EMBO reports. Thank you for your contribution to our journal.

In going through your manuscript prior to acceptance, I have noticed the identity of the bars and errors in supplementary figure 2 is not stated in the figure legend. Is it the mean +/- S.E.M. as the others? Please let us know and we will include this information.

We now encourage the publication of original source data -particularly for electrophoretic gels and blots, but also for graphs- with the aim of making primary data more accessible and transparent to the reader. If you agree, you would need to provide one PDF file per figure that contains the original, uncropped and unprocessed scans of all or key gels used in the figures and an Excel sheet or similar with the data behind the graphs. The files should be labeled with the appropriate figure/panel number, and the gels should have molecular weight markers; further annotation could be useful but is not essential. The source files will be published online with the article as supplementary "Source Data" files and should be uploaded when you submit your final version. If you have any questions regarding this please contact me.

As a standard procedure, we edit the title and abstract of manuscripts to make them more accessible to a general readership. Please find the edited versions at the end of this email and let me know if you do NOT agree with any of the changes.

As part of the EMBO publication's Transparent Editorial Process, EMBO reports publishes online a Review Process File to accompany accepted manuscripts. As you are aware, this File will be published in conjunction with your paper and will include the referee reports, your point-by-point response and all pertinent correspondence relating to the manuscript.

If you do NOT want this File to be published, please inform the editorial office within 2 days, if you have not done so already, otherwise the File will be published by default [contact: emboreports@embo.org]. If you do opt out, the Review Process File link will point to the following statement: "No Review Process File is available with this article, as the authors have chosen not to make the review process public in this case."

Thank you again for your contribution to EMBO reports and congratulations on a successful publication. Please consider us again in the future for your most exciting work.

Edited title and abstract

Chemerin15 inhibits neutrophil-mediated vascular inflammation and myocardial ischemia-reperfusion injury through ChemR23

Neutrophil activation and adhesion must be tightly controlled to prevent complications associated with excessive inflammatory responses. The role of the anti-inflammatory peptide chemerin15 (C15) and its receptor ChemR23 in neutrophil physiology is unknown. Here, we report that ChemR23 is expressed in neutrophil granules and rapidly upregulated upon neutrophil activation. C15 inhibits integrin activation and clustering, reducing neutrophil adhesion and chemotaxis in vitro. In the inflamed microvasculature, C15 rapidly modulates neutrophil physiology inducing adherent cell detachment from the inflamed endothelium, while reducing neutrophil recruitment and heart damage in a murine myocardial infarction model. These effects are mediated through ChemR23. We identify the C15/ChemR23 pathway as a new regulator and thus therapeutic target in neutrophil-driven pathologies.